# Non-Contact Human Vital Signs Extraction Algorithms Using IR-UWB Radar: A Review

**Zhihuan Liang** [1], **Mingyao Xiong** [2], **Yanghao Jin** [2], **Jianlai Chen** [2], **Dangjun Zhao** [2], **Degui Yang** [2], **Buge Liang** [2] **and Jinjun Mo** [1,*]

1    School of Information and Communication, Guilin University of Electronic Technology, Guilin 541004, China
2    School of Automation, Central South University, Changsha 410083, China
*    Correspondence: jinjunmo@guet.edu.cn

**Abstract:** The knowledge of heart and respiratory rates (HRs and RRs) is essential in assessing human body static. This has been associated with many applications, such as survivor rescue in ruins, lie detection, and human emotion detection. Thus, the vital signal extraction from radar echoes after pre-treatments, which have been applied using various methods by many researchers, has exceedingly become a necessary part of its further usage. In this review, we describe the variety of techniques used for vital signal extraction and verify their accuracy and efficiency. Emerging approaches such as wavelet analysis and mode decomposition offer great opportunities to measure vital signals. These developments would promote advancements in industries such as medical and social security by replacing the current electrocardiograms (ECGs), emotion detection for survivor status assessment, polygraphs, etc.

**Keywords:** impulse radio ultra-wideband (IR-UWB) radar; noncontact; short-range; vital signs





## 1. Introduction

Human vital signs, including cardiopulmonary signals, provide valuable parameters for the estimation of the physiological state of and the clinical reference for human beings [1,2]. There are quite a few applications in various fields for measuring these vital signs, for example, monitoring sleep conditions [3,4], for training processes [5,6], and other areas in combination with various technologies [7–9]. There are two approaches that manage to acquire these parameters: contact measurement based on physical ergonomics and non-contact monitoring based on radar signals. Wearable devices based on measurement techniques such as ECG, PPG, or heart pulses [10–14] still face certain limitations considering the various circumstances related to continuous monitoring. These can impact the subjects, causing inconvenience and placing a burden on certain groups such as the elderly, newborns, and patients with skin problems. Instead, contactless monitoring is capable of handling such problematic situations [15–17]. Furthermore, the contactless method of human monitoring also became essential during the COVID-19 pandemic [18], which caused reduced face-to-face exposure between subjects and system operators, such as patients and nurses.

Instead of obtaining the vital signals directly from the human body, non-contact monitoring takes advantage of radar signals to collect and evaluate the useful details of cardiopulmonary activities. Continuous-wave (CW) radar has been introduced for the remote monitoring of human subjects [19–21]. Frequency-modulated continuous-wave (FMCW) radar has also been applied by many researchers to measure the HR and the RR of a single target [22,23] or multiple targets [24] within a distance of a few meters. Moreover, impulse radio ultrawide-band (IR-UWB) radar has been introduced due to its capability of monitoring vital signalsin complex circumstances. For close distances, a type of wireless body area network (WBAN) [25] has been proposed using a UWB radar sensor for the detection of the respiration rate (RR) and the heart rate (HR) by side monitoring. For

relatively long distances, through-wall monitoring of patients for HR and RR estimation [26–28], their body movements [29,30], or also multiple targets [31,32] has been applied.

For the IR-UWB radar technology, the basic mathematical model describing the relation between the radar signal and human cardiopulmonary activities was introduced by [33, 34], which helps further develop the estimation of the RR and the HR by revealing the intermodulation product and harmonics of the respiration and heartbeat frequencies. To gain precise values of the RR and the HR, many algorithms have been introduced to process the extracted signals after various pre-treatments [35]. Thus, the performance of the proposed algorithms requires the verification of their estimation accuracy and real-time monitoring, as these two figures are essential for early warning and the timely response to a human subject who presents an urgent status, which could minimize possible injuries and save precious time for the intervention. This precious time saved would result from lessening the necessary processing time needed for and increasing the accuracy of the obtained data on the estimation of the patient's status, saving lives [36,37].

In this review, the algorithms used for the vital signals' extraction are presented and discussed, especially regarding their accuracy. The remainder of this review is organized as follows: Section 2 presents the basic structure of the radar system and the mathematical model of human cardiopulmonary activity monitoring. Section 3 explains the extraction of the cardiopulmonary signals and the evaluation standards for the algorithms, as well as the data used for processing. The results of each algorithm applied for the estimation of the HR and the RR are presented in Section 4.

## 2. The Monitoring Model Using IR-UWB Radar

This section includes the introduction to the structure of the UWB radar system and the monitoring of the target. The mathematical model of the human cardiopulmonary activity for the further processing of the received signals is introduced.

### 2.1. The Simplified Structure of the UWB Radar System

The simplified structure of the system, shown in Figure 1, mainly consists of two parts: the human–computer interaction terminal and the radar host. The human–computer interaction terminal works as a bridge between the user and the radar host, providing the commands and performing the transmission of the data. Moreover, the radar host consists of four main components: the receive antenna (RX), the transmit antenna (TX), the radio frequency (RF) transceiver front-end, and the control and processing system.

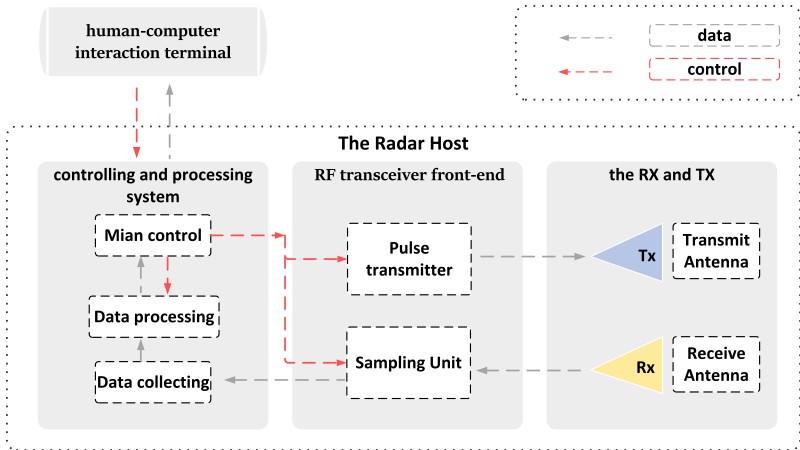

**Figure 1.** The simplified structure of the IR-UWB radar system.

The RX and TX were designed for radar signal reception and transmission, respectively. The RF transceiver front-end is mainly used to transmit electromagnetic pulses and to sample the radar echoes, and it includes the transmitter and receiver antennas. The

controlling and processing system performs the data collection and processing and hosts the main control system, which executes the commands input by the user.

The UWB radar system was developed by the School of Automation of Central South University. The trigger pulse from the pulse transmitter module is emitted by the TX antenna with a repetition frequency of 400 KHz. After being reflected, the signals are obtained by the RX antenna and sampled by the sampling unit module; then, they are converted into a digital signal by the analog-to-digital converter (ADC) and stored by the field-programmable gate array (FPGA) in the data collecting module. The parameters of the radar system are listed in Table 1.

**Table 1.** UWB radar system parameters.

| Parameter | Value |
|---|---|
| Center frequency | 500 MHz |
| Total bandwidth | 1 GHz |
| Sampling Points | 768 |
| Effective Range for vital signs monitoring | 8m max |
| Effective Pulse width | 2 ns |
| Pulse repetition frequency by equivalent sampling | 8 Hz |

### 2.2. The Monitoring of the Human Body

The human cardiopulmonary activities generate a series of chest wall movements, which change the distance between the radar antenna and the chest. This alteration could be detected based on the time-of-flight (ToF) of the radar signal. Typical human respiration ranges from 12 to 30 breaths per minute (bpm[1]) with a frequency band between 0.2 and 0.5 Hz and a chest movement amplitude of 0.5 to 1.5 cm. In addition, a typical heart rate ranges from 48 to 150 beats per minute (bpm[2]), corresponding to a higher frequency band between 0.8 and 2.5 Hz, arousing a chest motion amplitude of 2 to 3 mm. The human chest displacements could be extracted, considering the ToF of received radar echoes to estimate human vital signs. A mathematical model is introduced to reveal the exact procedures. Figure 2 shows that monitoring human cardiopulmonary activities could be presented as observing the ToF through $d(t)$. Two important parameters are the displacement amplitude of respiration $d_r$ and heartbeat $d_h$. In addition, $d_0$ is the chest-to-radar (CTR) distance.

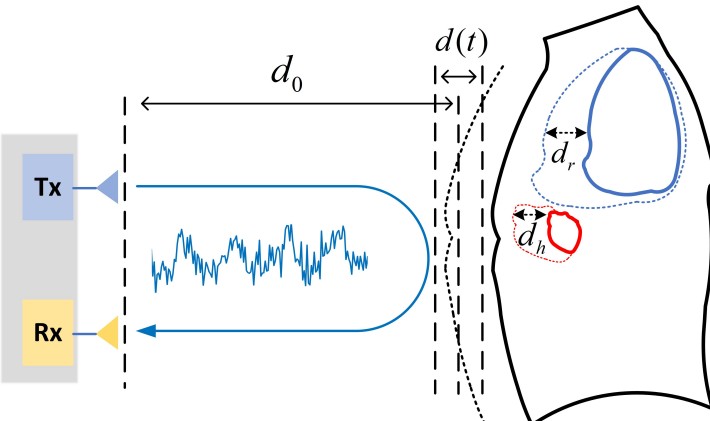

**Figure 2.** The monitoring of the human body.

Considering the chest wall as an infinite plane, the path the $m^{th}$ reflected wave travelled from the body could be presented as:

$$d = d_0 + d_r \sin(2\pi f_r t) + d_h \sin(2\pi f_h t)$$
$$t = mT_s$$

(1)

where $T_s$ is the pulse repetition interval, $f_r$ and $f_h$ are the frequencies of respiration and heartbeat.

Since there are three factors, the CTR distance and associated displacements of cardiopulmonary activities, responsible for the time delay, the ToF of the signal received could be expressed as:

$$\tau_d(t) = \tau_0 + \tau_r \sin(2\pi f_r t) + \tau_h \sin(2\pi f_h t) \tag{2}$$

where $\tau_0 = \frac{2d_0}{c}$ is the sum of the ToF between target and radar, $\tau_r = \frac{2d_r}{c}$ and $\tau_h = \frac{2d_h}{c}$ are delays caused by respiration and heartbeat displacements, and $c$ is the velocity of light.

In this situation, the received signal in fast time could be expressed as:

$$r(t, \tau) = \sum_{j=1} A_j p(t, \tau - \tau_j) + A_0 p(t, \tau - \tau_d(t)) \tag{3}$$

where $p(t, \tau)$ is the normalized pulse signal received, $A_j$ represents the amplitude of the received signal from the $j^{st}$ static objects and $\tau_j$ as its delay, $A_0$ is the amplitude of the received signal reflected from the target.

Then, the data matrix of radar echoes sampled could be expressed as:

$$R[m, n] = r(t, \tau = nT_f) \tag{4}$$

where $T_f$ is the sampling period in fast-time, and $n$ represents the sampling points of the signal in fast time.

## 3. The Materials and Evaluation Standards

This section introduces the vital sign signal extraction from the obtained data matrix and the evaluation standards for algorithm performances. In addition, some essential details of the measured and simulated data are explained.

### 3.1. The Extraction of the Required Signal

With the data matrix, obtaining the human vital signs signal then becomes a problem. One simple extraction approach is locating the maximum energy of each reflected echo. The calculation of energy using the data matrix could be expressed as:

$$E[n] = \sum_{m=1}^{M} |R[m, n]|^2 \tag{5}$$

where $M$ represents the total number of slow-time sampling points.

Then, the demanding section with maximum energy could be expressed as:

$$E_{\max}[k] = \max(E[i]), i = 1, 2, ..., N \tag{6}$$

where $k$ is the signal section required and $N$ represents the total samples in fast time.

The data selected $D_k$ for further processing could be expressed as:

$$D_k = R[m, k], m = 1, 2, ..., M \tag{7}$$

In addition, a schematic of the data matrix and selected $D_k$ is shown in Figure 3.

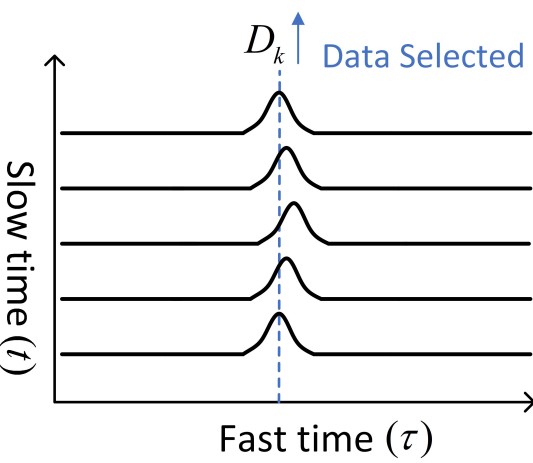

**Figure 3.** Data selected with maximum energy.

### 3.2. The Evaluation Standard

Different applications of vital signs extraction make requests for various algorithm performances. The accuracy of HRs and RRs is emphasized for assessments such as sleep quality and physical functional assessment of the elderly. Another factor, the real-time capability, which could be concluded as the monitoring interval for an algorithm to reach an acceptable accuracy, is demanded in scenarios such as emergency treatment and real-time vital signs monitoring during serious surgery. Under such conditions, this ability could be life-saving, with accurate vital signs being provided in time. Thus, the evaluation standards are organized as follows:

#### 3.2.1. Result Accuracy

To evaluate the performance of each algorithm, the accuracy of the RR and the HR becomes important. The resulting accuracy of each algorithm processing the same data section could be measured as follows:

$$Er = \frac{|D_{extract} - D_{real}|}{D_{real}} \times 100\% \tag{8}$$

where $D_{real}$ is the actual HR or RR of the target and $D_{extract}$ is the results obtained by the particular algorithm after processing.

#### 3.2.2. Real-Time Estimation Capability

HRs and RRs estimation usually include two major steps: obtaining a data matrix for a limited time and processing the data using an applied algorithm. Thus, the pre-response time before further body assessments separates into the monitoring interval and the processing time of one algorithm. Evaluations of the two parameters become essential since they challenge the real-time performance of each algorithm.

The typical human respiration frequency ranges from 0.2 to 0.5 Hz, corresponding to one cycle $T_r$ from 5 s to 2 s. Fortunately, all the algorithms have a processing time of fewer than 0.2 s, which is unavailable for a persuasive conclusion. In addition, it is negligible compared to the respiration cycle. Therefore, we consider the monitoring interval the primary factor in estimating the real-time capability. In addition, the acceptable estimation error of 10% is selected to verify it.

### 3.3. The Measured and Simulated Data

The four data are separated into three levels: $3T$, $2T$, and $T$. Furthermore, the standard interval $T$ is set to be 5 s to include at least one complete cycle of respiration and heartbeat. In addition, four groups of data, including measured and simulated ones, are chosen to make it comprehensive.

### 3.3.1. The Measured Data

Using the developed UWB radar mentioned in 2.1, the integrated system is 1.2 m above the ground. The experimental setups with a human subject are shown in Figure 4.

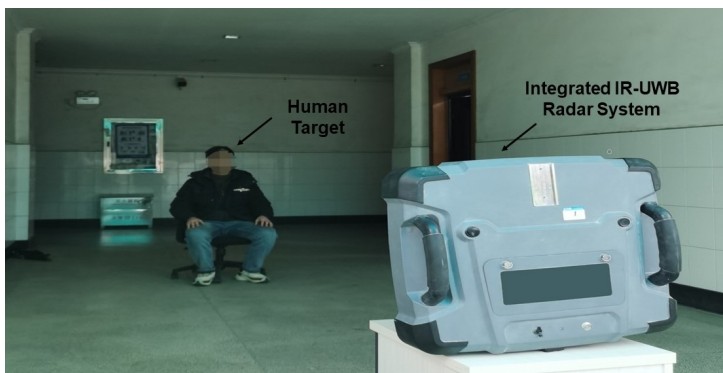

**Figure 4.** Experimental setup with human subject.

The HR and the RR of participants are obtained 6 m in front of the radar, sitting while monitoring, which is listed in Table 2. The monitoring of subjects is sustained for 30 s. These data are separated to suit the expected time levels, 5, 10, and 15 s.

**Table 2.** The Measured data.

| Items | Sections | RR (bpm[1]) | HR (bpm[2]) | Time Length (s) |
|---|---|---|---|---|
| Data 1 | Data 1 Section 1<br>Data 1 Section 2<br>Data 1 Section 3 | 19.8 | 76 | 5<br>10<br>15 |
| Data 2 | Data 2 Section 1<br>Data 2 Section 2<br>Data 2 Section 3 | 13.5 | 72 | 5<br>10<br>15 |

### 3.3.2. The Simulated Data

HRs and RRs are different between people groups, and they also vary under different situations. To make it comprehensive, the dataset must include extreme conditions for performance assessments. However, some HRs and RRs combinations require too much time and money for them to be acquired in the actual monitoring. Thus, the simulation compensates for such a situation.

The characteristics of the simulated data are listed in Table 3. The two simulated data are produced under the exact physical settings as measured ones.

**Table 3.** The simulated data.

| Items | Sections | RR (bpm[1]) | HR (bpm[2]) | Time Length (s) |
|---|---|---|---|---|
| Data 3 | Data 3 Section 1<br>Data 3 Section 2<br>Data 3 Section 3 | 30 | 48 | 5<br>10<br>15 |
| Data 4 | Data 4 Section 1<br>Data 4 Section 2<br>Data 4 Section 3 | 12 | 150 | 5<br>10<br>15 |

The uncommon combinations of HRs and RRs simulate the pathological characteristics caused by some severe diseases. One group includes acute respiratory distress syndrome (ARDS) [38], acute exacerbation of chronic obstructive pulmonary disease (AECOPD) [39], and acute heart failure (AHF) [40], with symptoms involving high respiration activity and

a decline in the HR comparing to an ordinary situation. Others, including viral myocarditis or chronic cor pulmonale, might lead to respiratory failure, arrhythmia, or overt heart failure [41,42]. Thus, the extreme values of the HR and the RR are in order to examine the performance of algorithms under such emergencies.

## 4. Algorithms and Results Discussion

This section introduces and compares various algorithms proposed by researchers for different vital signs extraction circumstances using IR-UWB radar. Some algorithms are improvements or modifications of other developed approaches applied early in other fields. Thus, comparing these algorithms with specialized ones for future research and application is necessary. A classification of introduced algorithms is managed to make it easier. To begin with, the brief structures of all algorithms are presented in Figure 5.

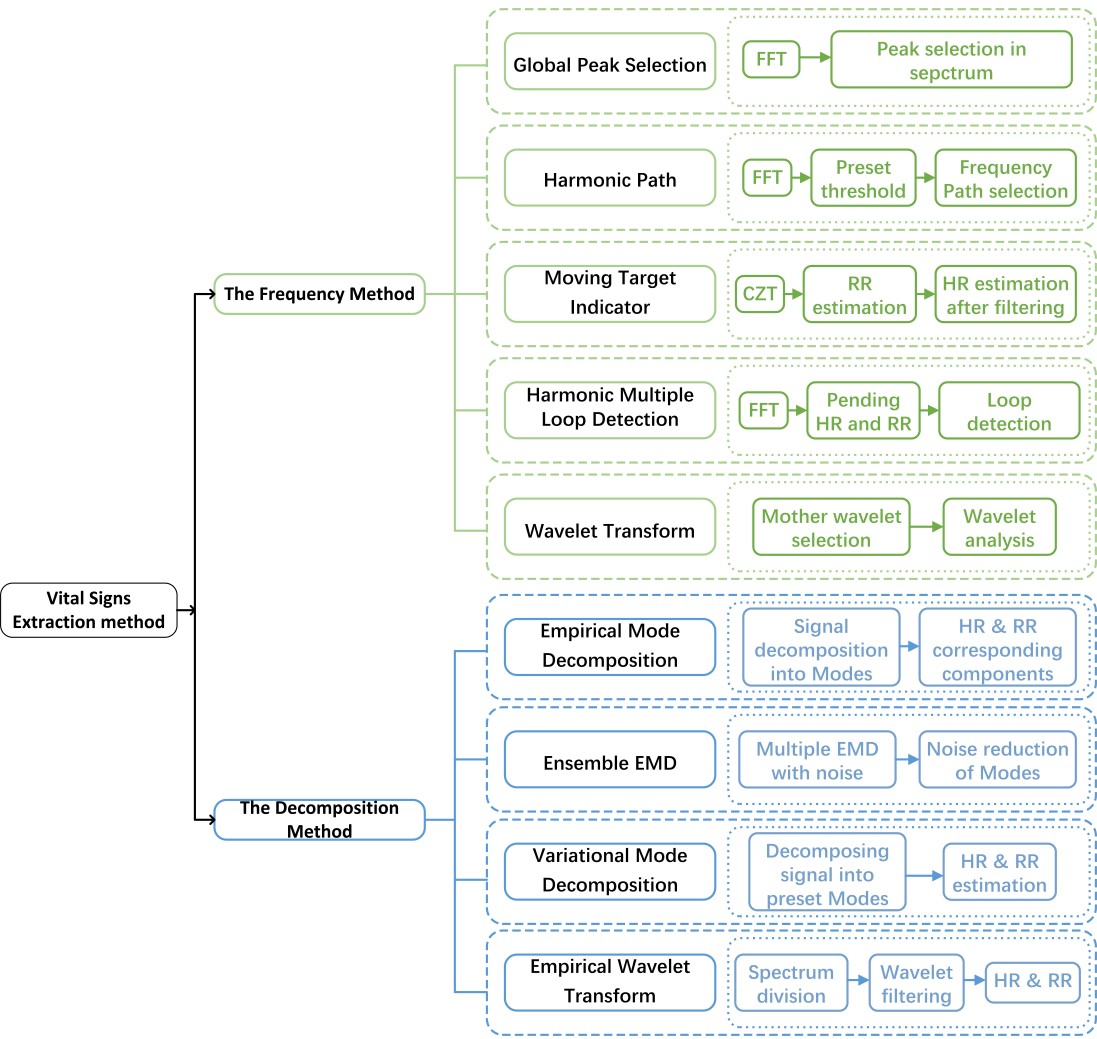

**Figure 5.** Brief introduction of vital signs extraction algorithms.

### 4.1. The Frequency Method (FM)

There are algorithms using frequency domain parameters to identify the HR and the RR components of targets using approaches such as fast Fourier transform (FFT) and chirp Z-Ttansform (CZT) [43] which can convert a signal from the time domain to frequency domain. The amplitude of the spectrum, even the linear relationship between the fundamental frequency and harmonics of heartbeat and respiration, is used to estimate HRs and RRs. Some typical algorithms are introduced and evaluated in this section.

### 4.1.1. Introduction of The FM Algorithms

The global peak selection (GPS) algorithm: Using the maximum magnitude of the spectrum after FFT, it asserts the HR and the RR in their corresponding frequency band. However, such an estimation might be strongly distorted by the harmonics and intermodulation products within the spectrum, leading to inaccurate values. This is demonstrated below, in section 4.1.2.

The moving target indicator (MTI)—CZT algorithm: To reduce the interference of such an effect after direct FFT of the $D_k$, the MTI technique is introduced to attenuate the respiration harmonics in the spectrum as the finite impulse response filter. In addition, to increase the accuracy of the operation, the CZT is applied to provide better resolution than the FFT in the selected frequency band without increasing sampling points [34]. The exact sequence of the algorithm uses the CZT to obtain a high-resolution spectrum and the maximum peak as RR. Then, the MTI operates and helps us to obtain a clear HR peak in the spectrum. Since there are no more relative RR components in the HR frequency band, there is no doubt that the performance is a promotion compared to the GPS.

The harmonic path (HAPA) algorithm: Another improvement is made in the HAPA algorithm, which realizes the HR and the RR estimation using the linear relation between fundamental and harmonic frequency of respiration and heartbeat [44]. The possible peaks of RRs and HRs with values above the preset threshold could generally form a path after FFT of $D_k$. In addition, the frequency estimation is based on the average inter-peak distance of the path. To make it available, we organize the principle of the preselected threshold of HAPA to be the same as in [44], which is 75% percentile of the obtained spectrum. However, it is also an issue in this principle, which might lead to no-path failure when forming a path using signals with interference. Unfortunately, the distortions happen in actual processing, with possible peaks ignored because of the threshold.

The harmonic multiple loop detection (HMLD) algorithm: Unlike the HAPA algorithm, the HMLD offers threshold-free processing and considers only two components, the fundamental and the second harmonics of HRs and RRs [45]. The HMLD algorithm initially selects the pending fundamental values of the RR and the HR after the FFT of $D_k$. Then, a reselection is issued if no pending second harmonics are corresponding. The benefits are obvious since the worries of suitable preset threshold disappear, and all ghost peaks are considered. Moreover, no problem exists with no-path concerns because of less distortion in the fundamental and second harmonic frequencies.

The wavelet transform (WT) algorithm: some researchers seek another time-frequency analysis approach, the WT, to realize vital sign estimation. Compared to Fourier transforms, the WT offers various wavelets for signal analysis, which allows researchers to find the most suitable mother wavelet to analyze the signal. However, the performance of all wavelets needs to be tested before application. The Meyer and Morlet wavelet is examined for the HR and the RR estimation [46,47]. Some relevant parameters of WT, the discretized wavelet scale, which is usually an estimated empirical value, the sampling frequency of $f_s = 2^3$, and the selected center frequency $f_0 = 2^8$ are settled. Considering the complex expression of the Meyer wavelet and its high similarity to the heartbeat in the waveform, the accuracy of using the Meyer wavelet might be superior to that of the Morlet wavelet at the HR estimation.

### 4.1.2. Processing Results Analysis

Figure 6 shows the spectrum of two vital sign signals extracted with a duration of 30 s with the same RR and HR as an example to verify harmonics interference. Figure 6a shows a distinct boundary between the RR and HR components due to a weak respiration activity. However, the situation is changed once vigorous respiration is involved. The HR estimation becomes complicated with relatively strong respiration, as shown in Figure 6b. The RR harmonic peak appears near the fundamental HR frequency, which might lead to false HR estimation. In addition, such a phenomenon is familiar and severe, considering various human body statuses and monitoring circumstances.

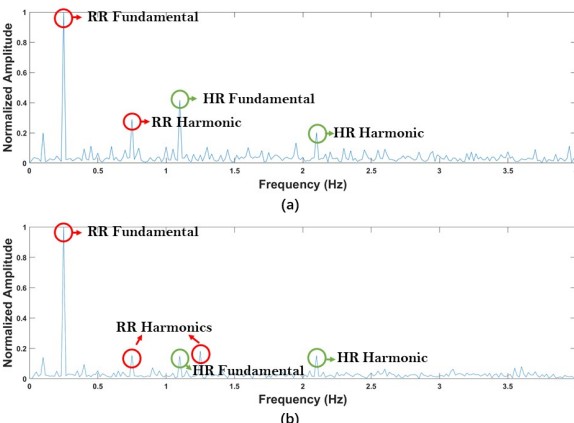

**Figure 6.** Spectrum comparison between different respiration situations. (**a**) Respiration with 1cm amplitude; (**b**) Respiration with 1.5 cm amplitude.

Thus, we first compare the processing results of each algorithm under extreme conditions to verify the performance using the simulated data. The HAPA algorithm faces no-path failures in both HRs and RRs estimation, as listed in Table 4. It has to be excluded from further FM algorithms comparison.

**Table 4.** The processing results of HAPA algorithm.

| Section | RR Error | HR Error |
| --- | --- | --- |
| Data 3 Section 1 | Failure | Failure |
| Data 3 Section 2 | 6.6% | 6.3% |
| Data 3 Section 3 | 6.6% | 8.3% |
| Data 4 Section 1 | 1.0% | Failure |
| Data 4 Section 2 | 16.7% | 0.0% |
| Data 4 Section 3 | 11.1% | 1.3% |

As shown in Figure 7, all FM algorithms have problems with data 3-processing, which reveals a challenge in close frequency analysis. Although many perform poorly except the MTI–CZT, the RR estimation seems acceptable with an error lower than 10% once with an extended data length: 2*T* and 3*T*.

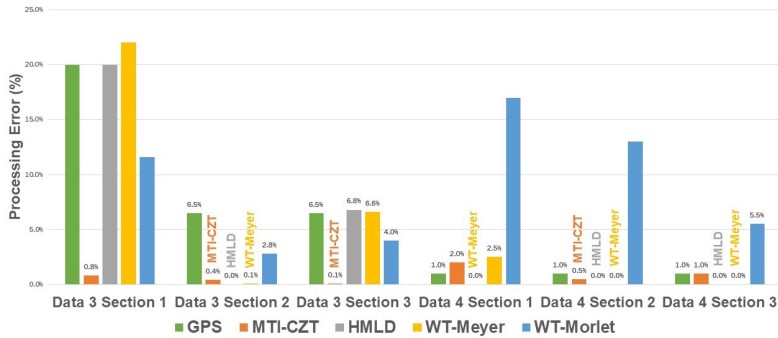

**Figure 7.** The simulated data processing error of different FM algorithms (RR).

However, the situation is altered when it comes to HR. None of them could separate the actual HR from the interference of RR harmonics in data 3 at an error under 10% according to Figure 8. Sadly, all FM algorithms are challenged by data 3-processing in HR estimation. Apart from data 3-processing, the HR results of data 4-processed by the GPS and the HMLD is also unsatisfactory, with errors of section 1 and 2 above 20%. However,

the GPS is excluded from further estimation for its even worse performance in processing section 3. On the other hand, the performance of some algorithms is acceptable in limited situations, such as the MTI–CZT and the WT–Meyer in data 4-processing.

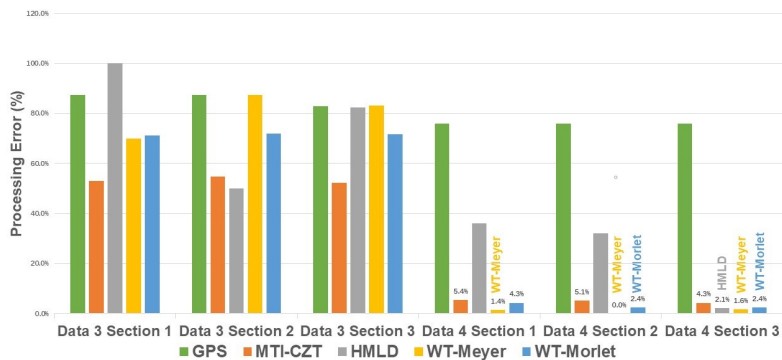

**Figure 8.** The simulated data processing error of different FM algorithms (HR).

The average error of these two algorithms in processing data 4 is listed in Table 5. Except for the two, the HMLD, with an average error of near zero in RR estimation, and the WT–Morlet, with an average error of 3.03% in HR estimation, stand out.

**Table 5.** The average error of data 4-processing.

| Algorithm | RR Error | HR Error |
| --- | --- | --- |
| MTI–CZT | 1.20% | 4.93% |
| WT–Meyer | 0.83% | 1.00% |

Moreover, the algorithm performance also requires verification in actual monitoring situations. With the results of the extreme test, we selected four FM algorithms for assessment: the MTI–CZT, HMLD, and WT algorithms with two wavelets. The processing results are shown in Figure 9, with numbers marked on those below 10% for clarity.

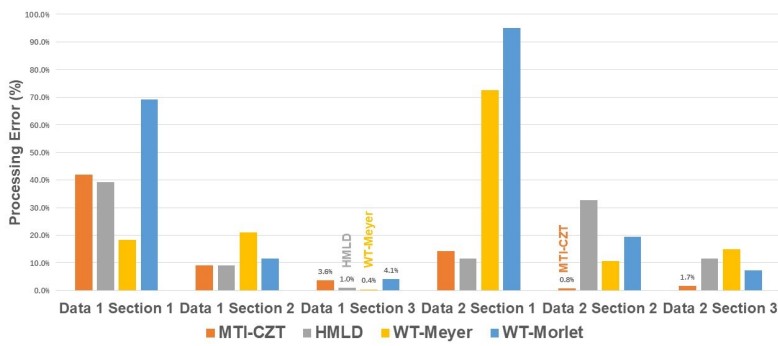

**Figure 9.** The measured data processing error of selected FM algorithms (RR).

The MTI–CZT has the best capability of obtaining high-accuracy RR results with limited time in section 2- and 3-processing. However, the FM algorithms struggle with accurate RR estimation using section 1 data with only one respiration cycle. Moreover, the RR estimation with the time length of 2*T* must be better to meet the required average error below 5%. In the HR estimation, the situation is slightly better. Despite a still unacceptable error in section 1 processing, the continuous observation of 10 s is adequate for HR estimation with tolerable error except for the HMLD algorithm when processing the signal 2 section 2, as shown in Figure 10.

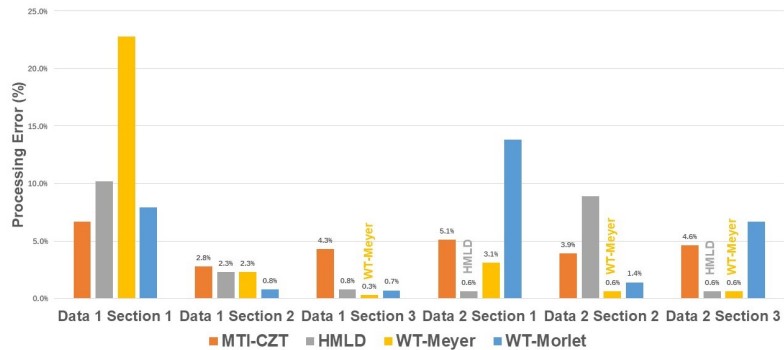

**Figure 10.** The measured data processing error of selected FM algorithms (HR).

The average error of these four algorithms processing 3*T* signal is listed in Table 6. The MTI–CZT algorithm has the best RR estimation accuracy as in other signal processing situations. In addition, the WT–Meyer algorithm has the best HR estimation accuracy once the samples are adequate.

**Table 6.** The average error of 15 s monitoring using FM algorithms.

| Algorithm | RR Error | HR Error |
|-----------|----------|----------|
| MTI–CZT | 2.65% | 5.90% |
| HMLD | 6.25% | 1.45% |
| WT–Meyer | 7.70% | 0.45% |
| WT–Morlet | 5.75% | 3.70% |

### 4.2. The Decomposition Method (DM)

Unlike the FM, the DM manages to separate the demanded HRs and RRs from obtained signals based on the characteristics of a signal itself. The DM algorithms decompose human vital sign signals into components containing respiration and heartbeat information, called intrinsic mode functions (IMFs) in some algorithms. The HR and the RR could then be reconstructed by analyzing the energy of obtained IMFs in the corresponding frequency band. Then, the HR and the RR frequency are available by applying the FFT to the reconstructed signal. This section introduces and evaluates some typical RRs and HRs extraction algorithms using IR-UWB radar.

#### 4.2.1. Introduction of the DM Algorithms

The empirical mode decomposition (EMD) algorithm: The EMD algorithm, proposed by Dr Huang in [48], is introduced for vital sign signal extraction in [49]. This paper applies the EMD algorithm for HRs and RRs estimation after determining the distance gate using Permutation Entropy. A critical advantage of the EMD is its adaptiveness. Calculating the average value of the upper and lower envelopes of a signal by its extreme points is an essential step of the EMD to obtain IMFs, depending only on the input signal. With $D_k$ as the input, the IMFs are obtained within seconds, just like other DM algorithms.

The ensemble empirical mode decomposition (EEMD) algorithm: The EEMD algorithm, an enhanced version of the EMD, solves the multi-mode problems of the EMD algorithm [50]. It was introduced in vital sign signal processing in [51]. Proper IMFs could be obtained with the added noise during the process. However, the introduced noise becomes another problem threatening the result accuracy. Thus, two different parameters play an essential role in the performance of the EEMD algorithm: the ensemble number of the EEMD and the ratio of the standard deviation of the added noise. The results face strong distortion once the ensemble number is not significant enough to eliminate the side effect of noise. A compromise must be made to balance the processing accuracy and the processing time in practice. In this paper, the particular ensemble number is set to 50, and the standard deviation of the added noise is set to 0.2, just as the exact parameters in [51].

The variational mode decomposition (VMD) algorithm: Another DM algorithm, the VMD [52] is also introduced into vital sign signal processing [26,53]. It has a parameter needed value pre-assignment, the number of intrinsic modes. This paper sets the exact value to 4, a practical value, as found in [26]. It combines the Wiener filtering, the Hilbert transform, and heterodyne demodulation to realize the decomposition. In addition, the VMD introduces a strict constraint upon the IMFs in [52], which differs from the EMD and the EEMD.

The empirical wavelet transform (EWT) algorithm: Unlike traditional wavelet transform requiring a mother wavelet, the EWT algorithm provides an adaptive wavelet transform approach [54]. It obtains the advantages of both adaptive decomposition algorithms such as EMD and the merits of the wavelet transform. The algorithm is introduced in vital sign signal processing in [30]. With $D_k$ as the input, the HR and the RR could be obtained by analyzing their sub-signals. A straightforward approach is to locate the corresponding peak in the spectrum.

### 4.2.2. Processing Results Analysis

The simulated data is introduced to examine the DM algorithms in extreme situations. In addition, the comparison of data 3-processing is emphasized, considering the less optimistic performance of the FM algorithms in all three sections. The processing accuracy of all four DM algorithms is shown in Figure 11. The estimation of RRs with long data length, 2*T* and 3*T*, is acceptable with error under 10% except for the VMD. In addition, the VMD and the EWT manage the RR estimation of section 1 with errors of less than 1%. As for data 4-processing, the preset modes number appears inappropriate, leading to poor performance of the VMD. As discussed, the same inadequacy happens in the EEMD processing, leading to an apparent accuracy decrease with the diminishing of samples. The processing error of the EEMD increases from 3% of section 3 to nearly 30% of section 1. However, the EWT performs steady RR estimation despite the conditions, with an estimation error of less than 5% in all data sections.

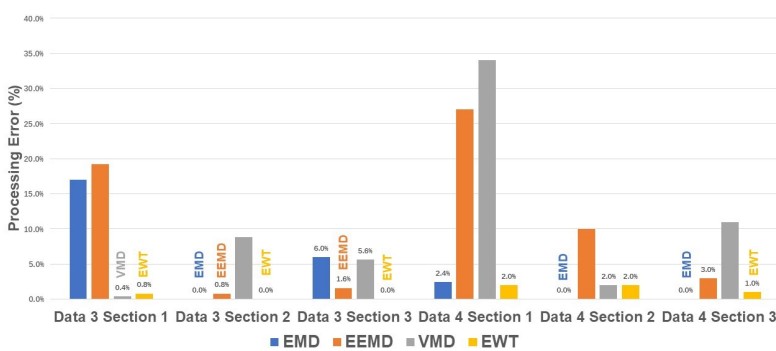

**Figure 11.** The simulated data processing error of different DM algorithms (RR).

Because of the strong distortion of closed RR frequency, the DM algorithms fail to realize accurate HR estimation of data 3, as the FM algorithms. The processing error of HRs is shown in Figure 12. However, the HRs estimation of data 4 reveals delightful accuracy. The EEMD, VMD, and EWT with average errors of 0.07%, 0.43%, and 0.3%, respectively, are all less than one $bpm^2$.

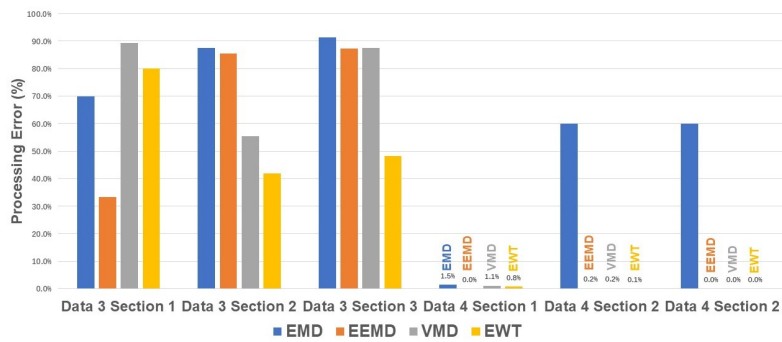

**Figure 12.** The simulated data processing error of different DM algorithms (HR).

Moreover, the algorithm performance also requires verification in actual monitoring situations. The results of RRs estimation are shown in Figure 13.

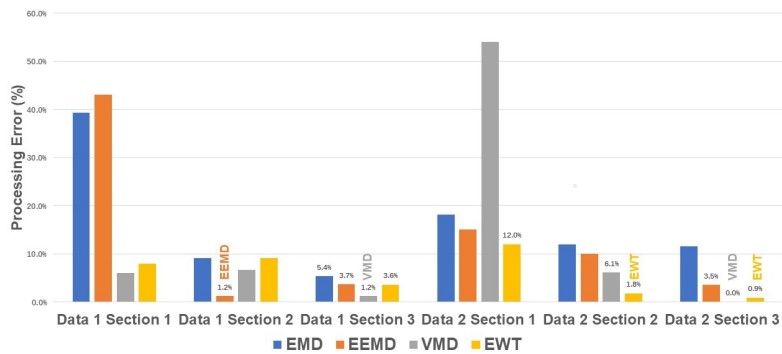

**Figure 13.** The measured data processing error of different DM algorithms (RR).

The accuracy of the EEMD seems unpleasant when processing 5 seconds of data with nearly 40% error in section 1 and almost 20% error in data 2. However, the performance of the EEMD improved since the monitoring time increased to $2T$, which becomes better than the accuracy of the EMD in RRs estimation. As for the VMD and the EWT, the estimation accuracy also reveals a pattern of increasing accuracy with augments of samples. This is also revealed in HRs estimation as shown in Figure 14.

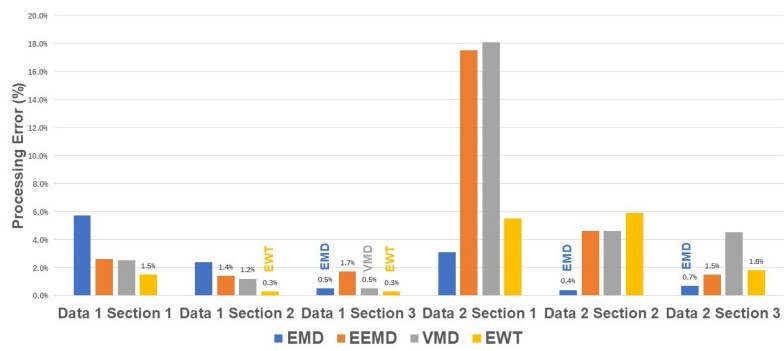

**Figure 14.** The Measured data processing error of different DM algorithms (HR).

All four algorithms have proved capable of HRs estimation in a normal cardiopulmonary condition with enough samples. None of them reaches a 10% estimation error. However, that situation altered once the samples decreased. The EMD, the EEMD and the VMD suffer inaccuracy in varying degrees. To further verify the performance of these algorithms, the average error of these four algorithms processing $3T$ signal is listed in Table 7.

**Table 7.** The average error of 15 s monitoring using DM algorithms.

| Algorithm | RR Error | HR Error |
| --- | --- | --- |
| EMD | 8.45% | 0.60% |
| EEMD | 3.60% | 1.60% |
| VMD | 0.60% | 2.50% |
| EWT | 2.25% | 1.05% |

The EEMD, VMD, and EWT algorithms could extract vital signs with errors below 5% if the time length of 15 s is acceptable. However, 15 s is hardly a real-time approximation. Thus, the real-time capability still needs to be examined.

*4.3. The Overall Analysis of Real-Time Capability*

Considering the previous failures in estimating close RR and HR situations, as in data 3, we select the appropriate one from the rest to assess the real-time capability of these algorithms. We choose measured data for analysis to gain conclusions close to typical circumstances. The outcomes of these algorithms with the average time spent to reach an estimation error of 10% are shown in Figure 15.

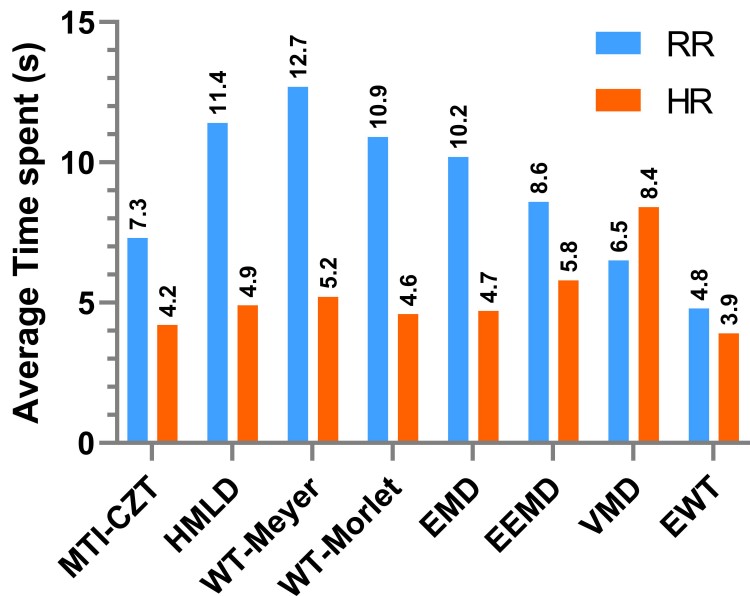

**Figure 15.** The average time spent on algorithms reach a 10% estimation error.

The values present the most capable algorithm, the EWT, with the time required for RR and HR estimation less than 5 s. Moreover, the VMD and the MTI–CZT show real-time ability on the RRs estimation. The WT–Morlet also performs well in the HRs extraction though the WT algorithm has a problem with RRs.

**5. Conclusions**

This paper comprehensively reviews the research and application of different non-contact human vital signal extraction methods. In addition, it presents the examination of algorithms using both measured and simulated data for simulating different monitoring situations. The processing accuracy and the time robustness are verified based on the results of each algorithm. Some reveal accredited performance in HR and RR estimation in given conditions. Unfortunately, many algorithms are unavailable for vital signs extraction under certain extreme conditions. Thus, further developments are required to manage these challenges and have accurate human vital signs extraction ability.

Moreover, the processing performance of particular algorithms in different vital sign extraction varies. Therefore, assembling multiple algorithms for better detection accuracy and system performance is a possible solution for promoting estimation performance. Furthermore, applying neural networks and feature detection for the pre-treatment or co-processing of the received data would also have the potential for accurate HR and RR estimation, which is also helpful for human vital sign extraction.

**Author Contributions:** Conceptualization, J.M. and D.J.; methodology, Y.J. and J.M.; software, Z.L. and M.X.; validation, Y.J., M.X. and Z.L.; formal analysis, Y.J.; resources, J.C., D.Y., D.Z. and B.L.; data curation, Z.L. and M.X.; writing—original draft preparation, Z.L.; writing—review and editing, J.M.,Y.J. and B.L.; visualization, J.C.; supervision, J.M. and J.C.; project administration, J.M. All authors have read and agreed to the published version of the manuscript.

**Funding:** This research received no external funding

**Informed Consent Statement:** Informed consent was obtained from all subjects involved in the study.

**Data Availability Statement:** The data that support the findings of this study are available from the corresponding author, J.M., upon reasonable request.

**Conflicts of Interest:** The authors declare no conflict of interest.

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
