# Peer review of "Non-Contact Human Vital Signs Extraction Algorithms Using IR-UWB Radar: A Review"

_electronics, doi:10.3390/electronics12061301_

Round 1

Reviewer 1 Report

This paper is interesting, but not informative due to lack of detailed explanation.

1. The description of the IR-UWB experimental device is insufficient.

2. Various algorithms have been presented, but the description of how they are implemented is lacking.

3. It is impossible to verify whether there is an issue in the implementation of the algorithm. Therefore, the authors compared and evaluated various algorithms, but this seems to be the authors' subjective evaluation.

Reviewer 2 Report

The study of heart and respiratory rates (HR and RR) has significant implications for various fields, including medical assessment, lie detection, and emotion detection. In this review, the authors aim to discuss the various techniques used for vital signal extraction from radar echoes and evaluate their accuracy and efficiency. The authors highlight the potential of wavelet analysis and mode decomposition in measuring vital sign signals and their potential to replace traditional methods like electrocardiogram (ECG) and polygraph in the medical and security industries. This review provides valuable insights into the development of techniques for vital signal extraction and the potential applications of these techniques.

The manuscript can be improved from the following aspects:

Line 26: Instead of obtaining the body signature directly from the human body. It should be “signal”.

Line 35: rate (RR) and heart rate (HR). The abbreviations names should be introduced the first time presenting.

English writing needs to be improved by native speakers. For example, 

Line 41: To gain a precise value of RR and HR, many algorithms are introduced to process the signal

extracted after various pre-treatments.  To gain precise values of RR and HR…

Line 54: should be “signals”

Line 65: RX and TX should be introduced by full names.

Line 76: Typical human respiration has 12 to 30 beats per minute. It sounds too high. Please find the literature to prove this conclusion.

Line 78: In the meantime, a normal heartbeat has a relatively high bpm range from 48 to 150. This cannot be the normal heartbeat range.

The axis and caption of the figures need to be addressed clearly.

Round 2

Reviewer 1 Report

The authors corrected all the points pointed out by the reviewer, so I consider this paper worthy of publication.